

# Prokaryote communities along a source-to-estuary river continuum in the Brazilian Atlantic Forest

Carolina O. de Santana[1], Pieter Spealman[2], Eddy Oliveira[3], David Gresham[2], Taise de Jesus[1] and Fabio Chinalia[4]

[1] Department of Exact Sciences (DEXA), Universidade Estadual de Feira de Santana, Feira de Santana, Bahia, Brazil
[2] Department of Biology, New York University, New York City, NY, United States
[3] Department of Biology (DCBIO), Universidade Estadual de Feira de Santana, Feira de Santana, BA, Brazil
[4] Institute of Health Sciences, Laboratory of Biotechnology and Ecology of Micro-Organisms, Institute of Health Sciences, Salvador, BA, Brazil

Corresponding author
Carolina O. de Santana,
cal.uefsbio@yahoo.com.br

## ABSTRACT

The activities of microbiomes in river sediments play an important role in sustaining ecosystem functions by driving many biogeochemical cycles. However, river ecosystems are frequently affected by anthropogenic activities, which may lead to microbial biodiversity loss and/or changes in ecosystem functions and related services. While parts of the Atlantic Forest biome stretching along much of the eastern coast of South America are protected by governmental conservation efforts, an estimated 89% of these areas in Brazil are under threat. This adds urgency to the characterization of prokaryotic communities in this vast and highly diverse biome. Here, we present prokaryotic sediment communities in the tropical Juliana River system at three sites, an upstream site near the river source in the mountains (Source) to a site in the middle reaches (Valley) and an estuarine site near the urban center of Ituberá (Mangrove). The diversity and composition of the communities were compared at these sites, along with environmental conditions, the former by using qualitative and quantitative analyses of 16S rRNA gene amplicons. While the communities included distinct populations at each site, a suite of core taxa accounted for the majority of the populations at all sites. Prokaryote diversity was highest in the sediments of the Mangrove site and lowest at the Valley site. The highest number of genera exclusive to a given site was found at the Source site, followed by the Mangrove site, which contained some archaeal genera not present at the freshwater sites. Copper (Cu) concentrations were related to differences in communities among sites, but none of the other environmental factors we determined was found to have a significant influence. This may be partly due to an urban imprint on the Mangrove site by providing organic carbon and nutrients *via* domestic effluents.

## INTRODUCTION

River ecosystems are frequently influenced by anthropogenic activities, which may lead to microbial biodiversity loss and/or changes in ecosystem functions and related services (*Mansfeldt et al., 2020*). Therefore, studies have been carried out to evaluate the significance of microbial community changes and how anthropogenic activities may influence such changes (*Reis et al., 2020*; *Zhang et al., 2020b*; *Lee et al., 2021*). However, since microbiomes remain unexplored in vast areas of the world, changes in sediment microbial communities of rivers are largely unknown at present, including in biomes that are under major threat.

One example is the Atlantic Forest extending along the Atlantic coast of South America, which is one of the most biologically diverse and most vulnerable biomes in the world (*Ministry of Agrarian Development of Brazil (MDA), 2010*). Human activities have drastically reduced the original cover of the biome, to only 11% of its pre-Columbian size on Brazilian territory (*Ribeiro et al., 2009*; *Silva & Nolasco, 2015*). One of the largest remaining fragments of the Atlantic Forest is located within the limits of the Pratigi Environmental Protection Area in the southern part of Bahia State, Brazil (*Ministry of Environment and Climate Change of Brazil (MMA), 2004*). Since its creation in 1998, the area has been subject to various environmental assessments, which have shown the effectiveness of the conservation efforts in the area (*Ditt et al., 2013*; *Lopes, 2011*; *Mascarenhas et al., 2019*), with the exception of a few local disturbances (*de Santana et al., 2021b*).

The aim of the present study was to determine the diversity and composition of bacterial and archaeal sediment communities along a tropical river in the Atlantic Forest of Brazil from the headwaters to the mouth. Given previously observed trends of decreasing microbial diversity along river lengths (*Wang et al., 2012*; *Behera et al., 2019*; *Zhang et al., 2020a*) and increasing human activity (*Statzner & Moss, 2004*) we hypothesized that the mangrove would exhibit the lowest biodiversity. However, we found that the mangrove site had levels of diversity comparable to the river source, potentially because of taxa, such as Archaea, that were unique to the site.

## MATERIALS AND METHODS

### Study area

Three sites were chosen along the Juliana River in the southeastern part of Bahia State, Brazil. The river drains the most important watershed in the region in terms of size and economic and ecological significance. Currently, the Juliana River is located entirely within a legally protected area, the Environmental Protection Area of Pratigi (Fig. 1). Its basin comprises an area of 299.8 km$^2$, through which the river runs almost linearly over 47 km. The source is in the Papuã Mountains. Several tributaries join the river along its way to the Serinhaém estuary (*Mascarenhas et al., 2019*; *Ditt et al., 2013*), where the city of Ituberá is located, a small urban area with less than 30,000 people where tourism is the main economic activity (*IBGE–instituto brasileiro de geografia e estatística, 2020*). Ituberá has been constructed within a mangrove forest, which has been retained along urban

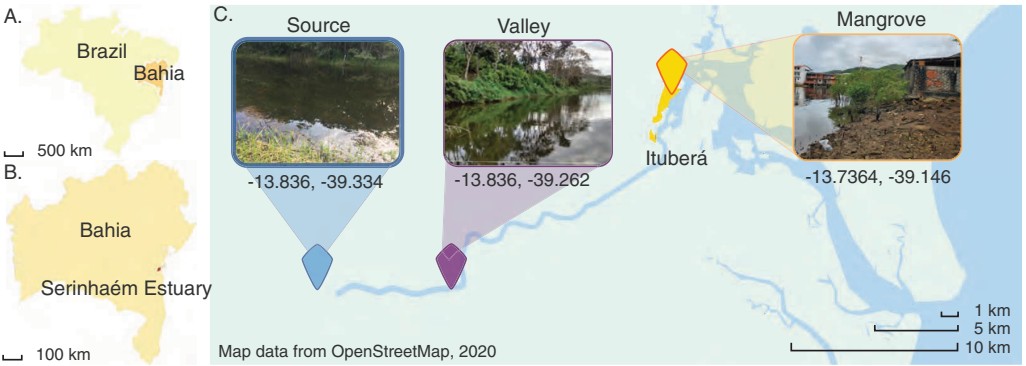

**Figure 1 Map of the Juliana River basin sediment sampling sites used in this study.** Map of the Juliana River basin and location and aspect of the three sites where sediment samples were taken. Map data from *OpenStreetMap (2020)*. Inset photographs taken by COS (*de Santana, 2020*).

waterways and mudflats (*de Santana et al., 2021b*). In contrast, most of the upstream reaches enable the observation of minimally impacted environments, because the upper portions of the watershed are considered to be highly conserved, lending themselves to ecological, hydrological and biogeochemical research. This includes studies of the biodiversity and ecology of microbial communities in mangrove sediments (*de Santana et al., 2021a*).

The Juliana River basin is subdivided into three administrative sections, I, II and III. Section I corresponds to the highlands of the Papuã Mountains. A site located there has been designated the Source site for the purpose of the present study. Section II corresponds to the downstream Valley region, which is mostly dominated by forest cover interspersed with a few agroforestry systems. Section III is the lowermost part of the hydrographic basin, hosting ecosystems ranging from tropical forest fragments to mangroves (*Mascarenhas et al., 2019*), including in and near Ituberá City close to where the sediments were collected. Nevertheless, this area still experiences little direct impacts by industrial development and family farming predominates land use (*da Silva Pereira et al., 2022*).

The field study presented here was approved by the state government organization Fundação de Amparo à Pesquisa do Estado da Bahia (project number: FAPESB/CNPq n° 794014/2013; permit number: 794014/2013). Portions of this text were previously published as part of a doctoral thesis (*de Santana, 2020*).

## Sampling and genomic analyses

Sediments were collected in February 2019 at the three sites selected in the Juliana River (Source, Valley, and Mangrove). At each site, three collection points were selected. Each site had to be at least 1.5 m apart from one another, free of visual vegetation, contamination, or pollution, and at a margin of the river where water depth exceeded 10 cm. Sediments were collected with a cylindrical core sampler, taking precautions to avoid disrupting rhizospheres associated with vegetation. Each sample consisted of a 10 cm deep surface sediment sample. Plant litter and other coarse particulate organic matter was manually removed from the core before placing the sediment samples in plastic bags on ice

in thermal boxes and immediately transporting them to the laboratory for chemical and genomic analyses.

Physical-chemical parameters such as temperature, pH, conductivity, and dissolved oxygen in the water column were measured at each site using a multiparameter probe (YSI model 85; Yellow Spring Instruments Inc., Yellow Springs, OH, USA). Additional environmental variables such as concentrations of Pb, Zn, Cu and Cd at each site have been previously reported (*da Silva Pereira et al., 2022*; *Mascarenhas et al., 2019*; Table S1). Since Cd concentrations were below detection limit at all sites, this variable was not included in the data analysis. In the laboratory, an aliquot of each sediment core was frozen at −20 °C for subsequent DNA extraction, while the remainder of the sample was used to measure organic matter (O.M.) content.

The total genomic DNA was extracted from 0.25 g of sediment using the PowerSoil DNA Isolation Kit (Qiagen, Germantown, MD, USA) and stored at −80 °C before analysis. After DNA extraction, the samples were sent on dry ice to Novogene Bioinformatics Technology Co. Ltd., Beijing, China for amplification of bacterial 16S rRNA genes, using the 515F and 806R primers (Table S2), followed by Illumina NovaSeq 6000 paired-end (2 × 250) sequencing (*Thompson et al., 2017*). Since sequencing of one of the samples from the Valley site failed, analyses were limited to the two remaining replicates.

Trimmomatic (*Bolger, Lohse & Usadel, 2014*) was used to filter and trim the demultiplexed sequences (ILLUMINACLIP:TruSeq3-PE.fa:2:30:10 LEADING:3 TRAILING:3 SLIDINGWINDOW:4:15 MINLEN:100). All reads were subsequently denoised using DADA2 (*Callahan et al., 2016*) in QIIME2 (*Bolyen et al., 2019*), merged using QIIME2 (File S1; Table S3), and then clustered into amplicon sequence variants (ASVs) (Table S4). Alpha-rarefaction was calculated using QIIME2 (Fig. S1) and set to 41,000 reads for the purpose of alpha- and beta-diversity analyses (Figs. S2, S3). All diversity analyses were performed using QIIME2's default parameters (File S1).

## Statistical analyses

Taxonomic assignment was performed using QIIME2's naive Bayes scikit-learn classifier (*Bokulich et al., 2018*) trained with the 16S rRNA gene sequences in the SILVA database (SILVA 138-99-515-806) (*McDonald et al., 2012*). The taxonomic feature table (Table S5) was resolved to the genus level for analysis (Table S6) using QIIME2. For each site, a bar chart was made of the phylum and class using the mean percentage of taxa abundance was calculated across replicates (Fig. 2A; File S2). Classes of high relative abundances (2% of the total community per site) and phyla were identified, and a heatmap of relative genus abundances generated for each replicate sample (Fig. S4; File S3).

Taxa resolved to the genus level were considered common across sites if they accounted for at least 0.1% of the reads per site, occurred in at least two replicates per site, or represented at least 1% of the reads in a single replicate. These criteria had to be met for each of the three sites (Fig. 2B; File S4; Table S7).

To determine how many taxa, resolved to the genus level, were only found at any given site, we first required each taxon to be minimally present at only one site. Minimal presence was defined as being greater than 0.001% of the total population per site, or being,

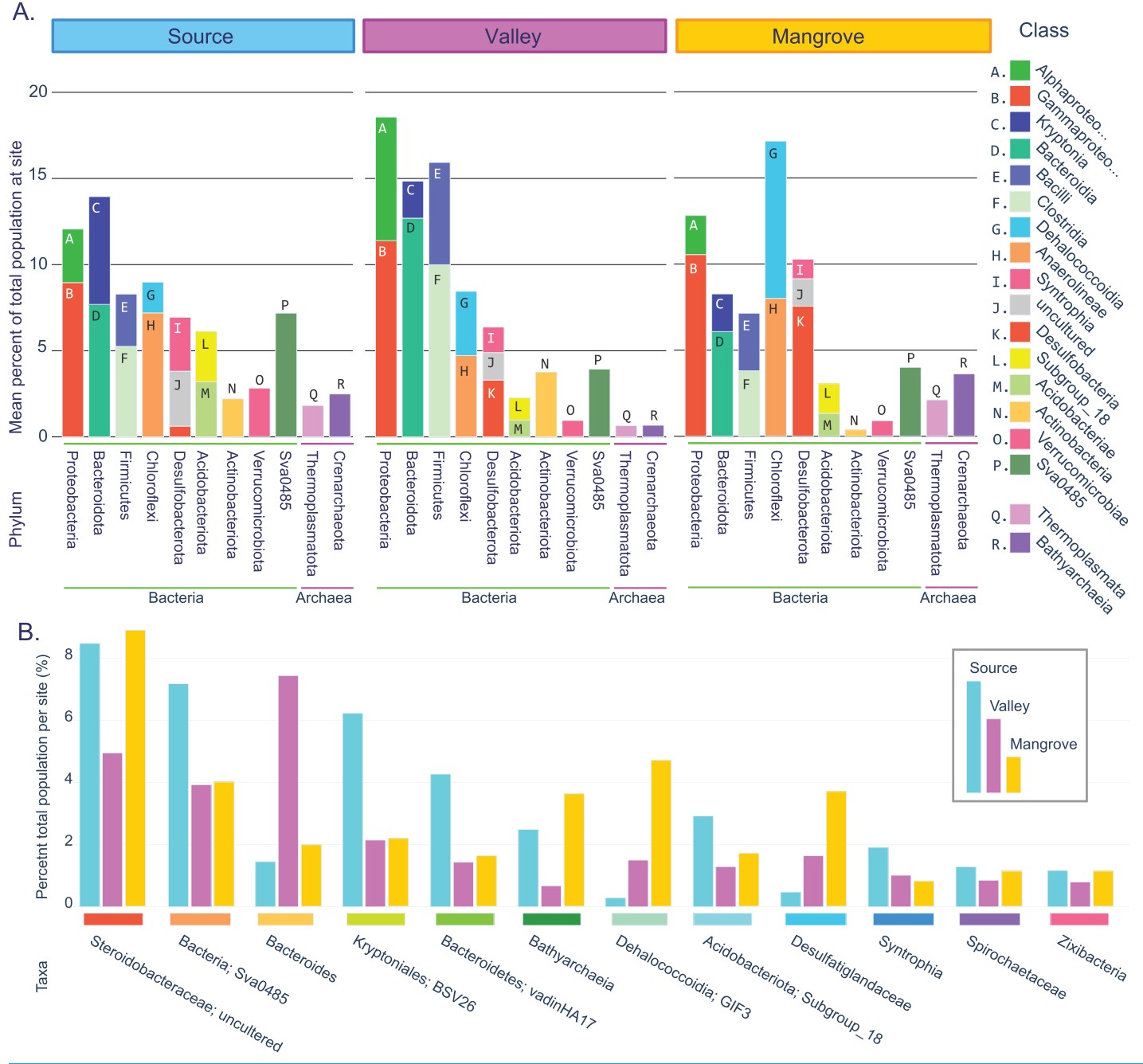

**Figure 2 Prokaryotic population statistics.** (A) Summary showing phyla and classes of all taxa accounting for an average of at least 2% of the prokaryotic community at least at one site. (B) Fifteen taxa that were highly abundant at all sites (>1% total per site).

on average, greater than 0.0001% of the population per site per replicate (Fig. S5; File S5; Table S8).

A site-specific analysis of significant differential abundances was performed using the ANCOM-BC package in QIIME2 (Tables S9–S11). We further subset these taxa to identify those that were significantly distinct to a single site, relative to the other two sites

(ANCOM-BC, q-value < 0.01) and that also represented a substantial percentage of the total population at that site (>1% total population), (File S6; Fig. 3D). A Venn diagram showing the overlap of significantly different taxa at each site is also available as a supplemental figure (Fig. S6).

The *Vegan* package (*Dixon, 2003*) was used to test correlations between community structure and environmental variables in R environment (version 4.2.2). Distances were calculated using metaMDS (distance used was Bray-Curtis) (Fig. S7; Table S12; File S7) and environmental variables were fit using envfit (Fig. 4B; Table S13; File S8).

The sequencing data is available from NCBI BioProject PRJNA650560. The entire computational workflow is available in a GitHub repository: https://github.com/pspealman/Project_Juliana_River_basin.

# RESULTS

## Taxonomic composition of sites and predominant groups

After quality filtering and taxonomic assignment, the 879,453 sequences remaining displayed the following pattern: 91.0% of the reads were associated with the kingdom Bacteria, 8.3% were associated with the Archaea and 0.6% were not assigned to either of these prokaryotic kingdoms. In total, ASVs were assigned to 85 phyla, 202 classes, 457 orders, 699 families, 1,089 genera and 458 species (Table S4).

We identified 18 highly abundant classes with a mean abundance per site of at least 2% (Fig. 2A). These classes constituted nine bacterial and two archaeal phyla. The two archaeal phyla, Crenarchaeota and Thermoplasmatota (as well Halobacterota, which was just below the 2% cutoff) were present at all sites, although they were most frequent in the mangrove sediments. For the Bacteria domain, the three sites shared similar dominant phyla, with Proteobacteria exceeding 10% and Bacteroidota, Bacillota (Firmicutes), Chloroflexota, and Desulfobacterota accounting each for >5% at all sites. Combined, these five phyla and their 11 classes represented the majority of the prokaryotic populations (50–64%) at each site.

This large overlap prompted us to assess how many of the more abundant genera were present at all sites (see Methods). We found 87 such taxa, 77 of which were resolved to the genus level (Table S7; Fig. 2B), which together accounted for 72% (Source) and 61% (Valley and Mangrove), respectively, of the total abundance and could thus constitute the core microbiome in sediments of the river.

## Community differences among sites

ANCOM-BC analysis indicated that abundances of numerous taxa significantly differed between pairs of sites (Figs. 3A–3C; File S6). The greatest difference occurred between the Source and Mangrove sites (Fig. S6; Tables S9–S11). Genera specific to only one of the study sites (Fig. S5) included 87 taxa that were unique to the Source site, two to the Valley site, and 63 to the Mangrove site. However, these taxa represent very small proportions of the total communities, with 0.65% being unique to the Source site, 0.03% to the Valley and 1.1% to the Mangrove site (Table S8). Resolved to the genus level, some taxa were significantly more abundant at one site compared to the two others (ANCOM-BC, q-value < 0.01) and represented a notable percentage of the total abundance at that site

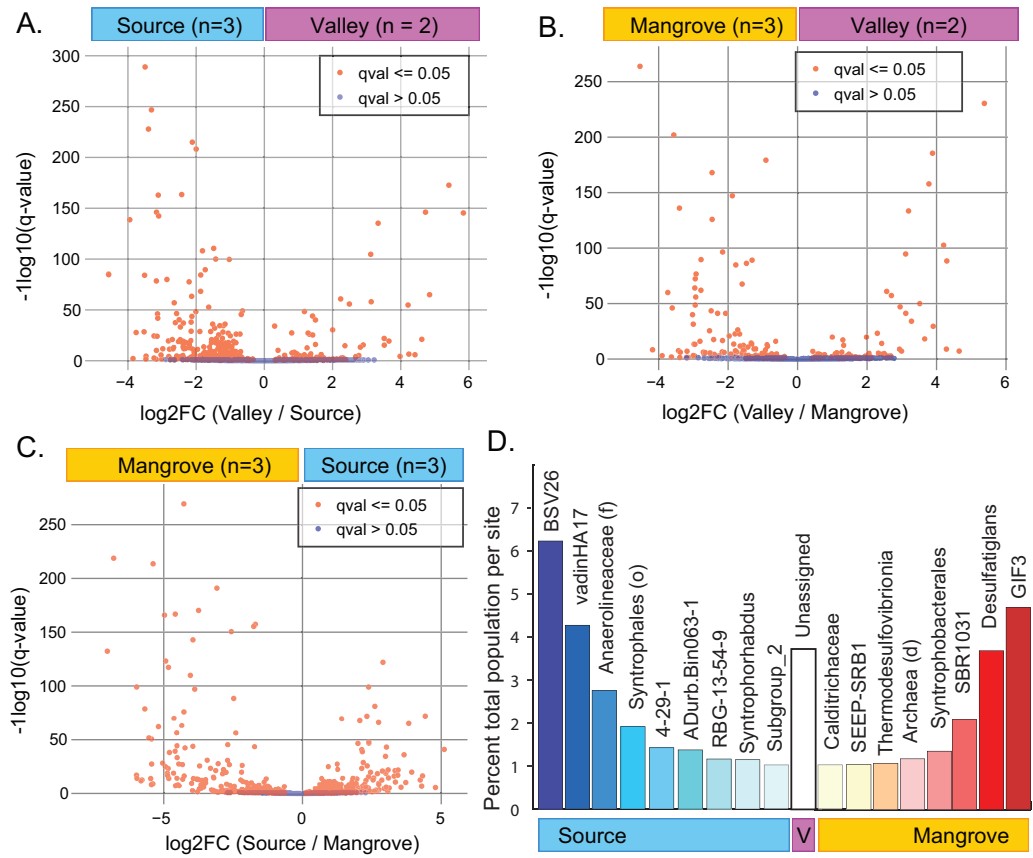

**Figure 3 Results of differential abundance analysis using ANCOM-BC.** (A–C) Indentification of differences in the abundance of taxa (down to the genus level) between pairs of sites. (D) Subset of taxa at each site (down to the genus level of) that were distinct to that site and represented a substantial percentage of the total abundance (>1%).

(>1% total population). We found nine such taxa at the Source site and eight at the Mangrove site (Fig. 3D), whereas none were more abundant at the Valley site, although sediments at that site had more reads that could not be assigned to any taxon ('Unassigned').

## Community structure, diversity and environmental variables

Prokaryotic diversity expressed as the Shannon entropy index was highest at the Mangrove and lowest at the Valley site (Fig. 4A); however, site differences were only significant in the omnibus test ($p = 0.04$). Similarly, differences in community composition between sites assessed by the Weighted UniFrac distance measure (Fig. S3) were only significant in the omnibus PERMANOVA ($p = 0.007$). Site differences among the prokaryotic communities are also shown in the PCA, which separated the Source site from the Valley and Mangrove sites along PC1 (Fig. 4B), with copper (Cu) concentration as the most influential environmental variable ($p = 0.011$). Nearly significant differences in the concentration of zinc (Zn) ($p = 0.063$) were primarily related to PC2, whereas temperature, dissolved oxygen, organic matter (O.M.), Ni, salinity, Cr, pH, and Pb had no significant influence.

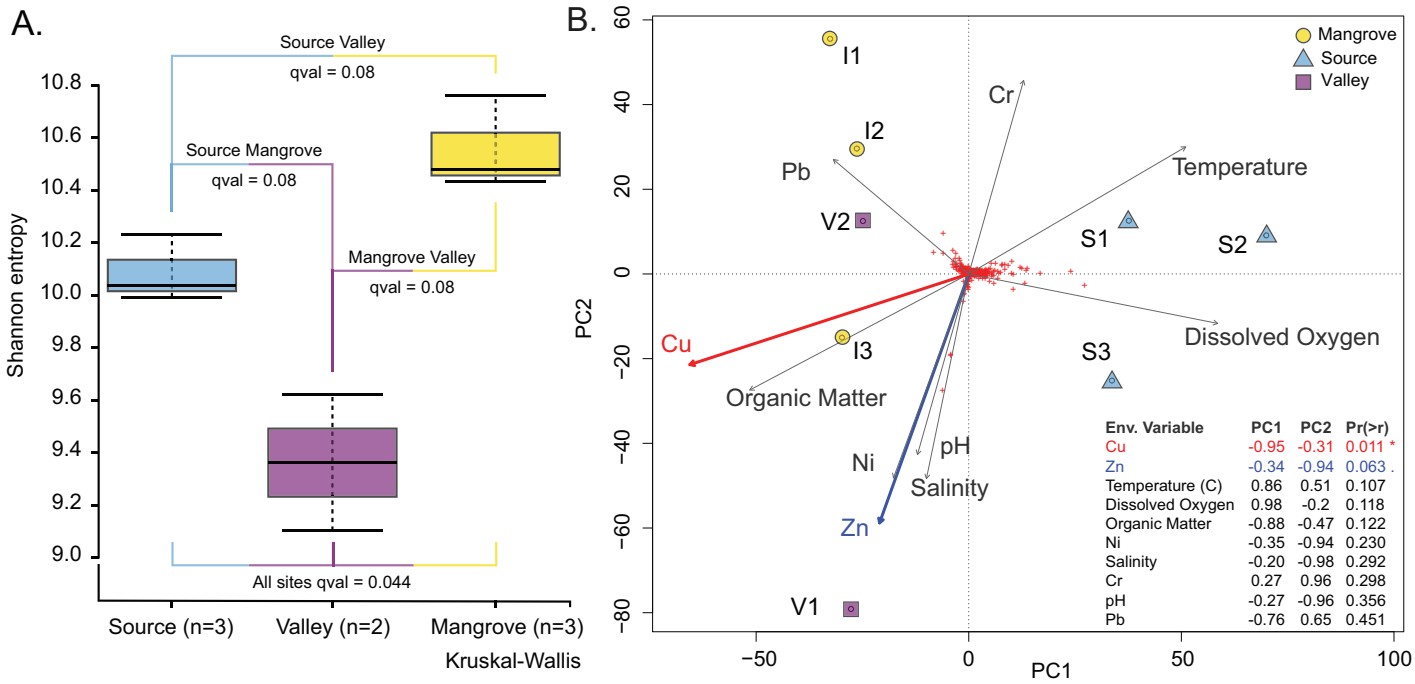

**Figure 4 Prokaryotic population characteristics.** (A) Shannon alpha-diversity indices of prokaryote communities at the Source, Valley and Mangrove sites. (B) PCA plot relating sediment prokaryote community composition to environmental variables at the three sites.

# DISCUSSION

Our results suggest a shift in prokaryote diversity along the river continuum from the headwaters (Source) to the mouth (Mangrove), with a minimum occurring in the middle reaches (Valley). One potential reason for the decrease from the headwaters to the middle reaches could be increasing anthropogenic influences, including contamination, as seen in previous studies (*Berg et al., 2012*; *Chen et al., 2018*). However, given the conservation status of the Julian River and the limited number of sites and samples in the present study, this tentative conclusion remains speculative, since a range of other factors may have influenced the prokaryotic sediment communities. Moreover, given the differences observed in both communities and environmental variables at the Mangrove site, it remains unclear to what extent the increase in diversity at this urban site was due to factors not measured in our study, including local anthropogenic impacts.

Previous studies of sediment microbial communities along river-estuary continua have found a decreasing trend of microbiome diversity in the direction of the river flow (*Wang et al., 2012*; *Behera et al., 2019*; *Zhang et al., 2020a*; *de Santana, 2020*). Variables such as temperature, salinity and trophic state were strongly related to the taxonomic and functional composition of microbial communities in those studies, in contrast to the present study where only Cu concentrations were significantly related to differences in the prokaryotic communities among sites.

Diversity is expected to decrease with increasing habitat harshness (*Statzner & Moss, 2004*), which is frequently associated with environmental disturbances. Accordingly, we expected the community in our mangrove sediments to be less diverse than the freshwater sediments, but we observed the opposite trend in that the mangrove site displayed the highest prokaryotic diversity. Considering that environmental conditions in mangrove sediments differ fundamentally from characteristics at freshwater sites, prokaryote diversity is expected also to differ between those sites. Additionally, wastewater discharge may have an influence by supplying organic matter and nutrients in readily accessible forms, which may override adverse effects of habitat harshness on prokaryotic diversity (*de Santana et al., 2021b*).

Gammaproteobacteria were well represented within the phylum Proteobacteria, including an uncultured genus in the Steroidobacteraceae that was both common across sites and frequent. While members of the Steroidobacteraceae family have been recognized as key taxa in aquifers (*Abiriga, Jenkins & Klempe, 2022*) and in association with Rhizobiales in plant rhizospheres (*Sakai et al., 2014*), this uncultured genus may occupy a similar, but different, niche. Presence of the phylum Bacteroidota in sediments has been related to environmental characteristics such as trophic state and temperature (*Huang et al., 2017*; *Dai et al., 2016*), suggesting that resource availability and environmental conditions were conducive to this group along the river continuum. Another highly abundant phylum was Sva0485. Recently reported but not well characterized, this group is often a member of sulfate-reducing assemblages where it is thought to play an important role in the sulfur cycle of freshwaters (*Chen et al., 2023*).

The prevalence of Proteobacteria and Firmicutes in the sediments of all our study sites is in general agreement with literature reports from soils and sediments (*Tveit et al., 2013*; *Jost, 2007*; *Yadav et al., 2015*; *Andreote et al., 2012*; *Imchen et al., 2018*; *Su et al., 2018*) and has been ascribed mainly to the high morphological and physiological diversity of these groups that enable the colonization of diverse habitats. However, aside from the majority of generalists, we also found some level of site-specificity, with some taxa showing preference and even exclusivity for the Source, Valley or Mangrove sites. In general, we found preferences for the Mangrove site for groups which are prevalent in coastal environments, such as the archaeal phyla Thermoplasmata, Halobacterota, and Crenarchaeota (*Thiele et al., 2017*). Many of the characterized groups of Crenarchaeota are thermophilic, have a preference for anaerobic environments, such as sediments, and may also be acidophilic (*Leigh & Whitman, 2013*; *Shakir et al., 2023*). While mangroves are often characterized as alkaline (*Caldeira & Wickett, 2003*) previous isolation of acidophilic fungi (*Gao et al., 2020*) suggests that this generalization may not hold for microbiomes, especially at anthropogenically impacted sites. Halobacteridota are known to succeed in environments with high salt concentrations and the genera we found exclusively at the Mangrove site are highly correlated with methanogenesis (*Yang et al., 2022*). While possibly a product of urban runoff (*Li et al., 2019*), this is also consistent with our increasing understanding of the role of methanogenesis in mangroves (*Hu, He & Wang, 2024*). Overall, these results suggest that while some taxa are broadly distributed in

sediments along the river continuum, many of the taxa we identified survive in specific environmental conditions.

The majority of the 88 taxa unique to the Source site belonged to the Bacteria domain, with two genera of the methanogenic archaeal phylum Halobacteridota. From the bacterial groups, we found taxa with varied importance in ecological, biotechnological and in human health contexts, such as *Methylocystis*, a methane-oxidizing genus that has been studied for the purpose of mitigating methane emissions, and *Anaerococcus*, which are anaerobic species commonly found in human microbiota (*Dedysh, Knief & Dunfield, 2005*; *Murphy & Frick, 2013*). The family Sporolactobacillaceae and the genus *Microbacterium* were exclusively found in the sediments from the Valley. While *Microbacterium* is known to be quite widespread and common in a variety of environments (*Evtushenko & Takeuchi, 2006*), the endospore-forming Sporolactobacillaceae are primarily known from food spoilage and biotechnological systems (*Harirchi et al., 2022*).

Several taxa were associated with anaerobic biodigestion, including vadinHA17 in the Bacteroidetes (*Zhou & Xu, 2020*), ADurb.Bin063-1 in the Pedosphaeraceae (*Gaio et al., 2023*), and Anaerolineaceae (*Yamada & Sekiguchi, 2018*), consistent with the observation that the water at the Source site had the lowest dissolved oxygen concentrations (Table S1). While several taxa we found are considered sensitive to heavy metals, including 4-29-1 which belongs to the Nitrospirota (*Wang et al., 2022a*) and ADurb.Bin063-1 (*Chun et al., 2021*), we also found taxa resistant to trace metals, such as *Syntrophorhabdus* (*Da Costa et al., 2023*) and Subgroup 2 (GP2) of the Acidobacteriota (*Wang et al., 2022b*). Notably, GP2 has previously been found to be significantly associated with undisturbed tracts of the western Amazon rainforest (*Navarrete et al., 2015*) and the Atlantic Forest (*Catão et al., 2014*), consistent with the conservation status of the Juliana river basin.

Conversely, we found the Mangrove site to be enriched in several genera associated with disturbed ecosystems. These include *GIF3* (Dehalococcoidia) observed to rapidly arise in sediments of disturbed riverbanks (*López-Lozano et al., 2013*), and *Desulfatiglans*, a potential polycyclic aromatic hydrocarbon (PAH) degrader in urban rivers (*Li et al., 2022b*). Furthermore, both *Desulfatiglans* and *SEEP-SRB1* (Desulfobacterota) are associated with urban mangroves with high sulfate ($SO_4^{2-}$) and iron (Fe) concentrations and low nitrate ($NO_3^-$) and P (*Li et al., 2022a*) concentrations. *SEEP-SRB1* is also a syntrophic sulfate-reducing bacterium (SRB) capable of anaerobic methane oxidation (AOM) in obligate partnership with anaerobic methanotrophic archaea (ANME) (*Murali et al., 2023*). This could suggest a potential relationship with some of the unassigned Archaea observed at the site. However, many distinct environmental factors may contribute to the investigated mangrove being the most different site in the present study, especially because of the coastal tidal environment, in addition to its urbanization.

## ACKNOWLEDGEMENTS

The authors would like to thank the Organização de Conservação da Terra (OCT) for providing infrastructure for the field work in the environmental protection area.

### Funding

This study was financed by the Coordenação de Aperfeiçoamento de Pessoal de Nível Superior–Brasil (CAPES)–Finance Code 001 and by the Fundação de Amparo à Pesquisa do Estado da Bahia–Fapesb. There was no additional external funding received for this study. The funders had no role in study design, data collection and analysis, decision to publish, or preparation of the manuscript.

### Grant Disclosures

The following grant information was disclosed by the authors:
Coordenação de Aperfeiçoamento de Pessoal de Nível Superior–Brasil (CAPES)–Finance Code 001.
Fundação de Amparo à Pesquisa do Estado da Bahia–Fapesb.

### Competing Interests

The authors declare that they have no competing interests.

### Author Contributions

- Carolina O. de Santana conceived and designed the experiments, performed the experiments, analyzed the data, prepared figures and/or tables, authored or reviewed drafts of the article, and approved the final draft.
- Pieter Spealman performed the experiments, analyzed the data, prepared figures and/or tables, authored or reviewed drafts of the article, and approved the final draft.
- Eddy Oliveira conceived and designed the experiments, authored or reviewed drafts of the article, and approved the final draft.
- David Gresham analyzed the data, authored or reviewed drafts of the article, and approved the final draft.
- Taise de Jesus conceived and designed the experiments, authored or reviewed drafts of the article, and approved the final draft.
- Fabio Chinalia conceived and designed the experiments, performed the experiments, analyzed the data, authored or reviewed drafts of the article, and approved the final draft.

### Field Study Permissions

The following information was supplied relating to field study approvals (*i.e.*, approving body and any reference numbers):

Field experiments were approved by the governmental organization FUNDAÇÃO DE AMPARO À PESQUISA DO ESTADO DA BAHIA (project number: FAPESB/CNPq n° 794014/2013).

### DNA Deposition

The following information was supplied regarding the deposition of DNA sequences:
All sequencing data is available from NCBI: PRJNA650560.

## Data Availability

Data and code, such as the shell commands used for QIIME2 or secondary analysis scripts in python or R, are available in the Supplemental Files.

## Supplemental Information

Supplemental information for this article can be found online at http://dx.doi.org/10.7717/peerj.17900#supplemental-information.

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
