# Peer review of "Prokaryote communities along a source-to-estuary river continuum in the Brazilian Atlantic Forest"

_PeerJ, doi:10.7717/peerj.17900_

## Round 0.1 · original submission · Major Revisions

I have now received two cogent reviews of your manuscript, whose assessments perfectly concur. As you will see below, both identify major problems (e.g., the use of PiCrust to infer functional features, lack of coherence) in addition to making numerous other thoughtful comments and suggestions that will be useful to strengthen the manuscript. Please give particular attention to the main issues when revising the manuscript in the light of these comments.

I look forward to receiving the revised manuscript in due course. I will then send it out for review again before making a final decision. In your cover letter please state point by point how you have addressed the reviewers' criticism and suggestions.

**Language Note:** The review process has identified that the English language must be improved. PeerJ can provide language editing services - please contact us at [email protected] for pricing (be sure to provide your manuscript number and title). Alternatively, you should make your own arrangements to improve the language quality and provide details in your response letter. – PeerJ Staff

Reviewer 1 ·

Basic reporting

General:

1. Detailed information of applied wet lab protocols (e.g. PCR settings), statistical tests, and bioinformatical analyses would be useful to follow the workflow. Metadata of sampling points/time would be needed.

2. Introduction and discussion are very detailed with a lot of background.

Detailed:

Line 28 -34: Very general introduction to the importance of microbes in ecosystem. Since this manuscript is focused on river ecosystems, a more focused background on microbes in stream sediments may be more suited.

Line 40: Repetition of “of”

Line 41: Maybe it would be useful to mention here how many sampling sites were considered, i.e. the different biomes which were covered.

Line 43: Here and in the following, the authors often use the term “16S rRNA”. Since this analysis is based on DNA sequencing, it would be applicable to write “16S rRNA gene”. Where was quantitative 16S rRNA analysis performed?

Line 44/45: “other sites” is very general. If the other sites are mentioned above, one could refer here to these specific sites.

Line 50: It is unclear to what “family clusters” refer to: A taxonomic family or a functional KEGG family.

Line 44-54: The authors mention “significant” difference, “trends”, “distinct” microbiomes. Therefore, adding for example p-values or other indices to these statements would confirm these.

Line 52-54: “Our results confirm that prokaryotic communities are composed of significantly distinct populations”. While I agree that the authors analysis leads to this statement for their analyzed ecosystem, it is difficult to extrapolate to all prokaryotic communities in general. Thus, I would suggest to get more specific here and focus on the analyzed ecosystem.

Line 65: “A large number of studies”. Very broad term.

Line 67-68: Please, check back if the citation style is correct. Same for other citations with multiple references are within one sentence.

Line 72: “In Brazilian territory, the original cover of Atlantic Forest has been drastically
73 reduced to only 11% of its pre-Columbian size.” Please add a reference for that.

Line 81: “the most important watershed in its region”. Please add which in which regard it is important, e.g. economically, ecologically.

Line 82: “29.975 ha”. Please transform this in a SI unit.
Line 106: Could the authors please specify if the water in section III, i.e. their sampling point “mangroves”, is only fed by water coming from the other sampling points. Otherwise, one would need to reconsider if it is really a “continuum” as stated in the title.

Line 182: “Prokaryotic diversity was determined using total DNA analysis of 16S rRNA.” Please, rephrase this sentence. Since only a fragment of the 16S rRNA gene was amplified it is not correct that the total DNA was analyzed.

Line 219: “In agreement with the diversity trends previously described,” Please, add where.

Line 324: “As we confirmed in initial analysis”. Please, add where.

Regarding references: The citation style is not uniform, e.g. sometimes a DOI is added.

Fig1: Please, add a broader picture of the map. For example, show where the basin is located in the country, continent.

Fig2: As I understand it, you used a NMDS and not a PCA, as shown in the figure description. The stress value is missing for the NMDS. Is the third replicate “V3” missing in the NMDS?
Regarding C/D: Please consider something like a volcano plot to show the different taxa.

Experimental design

General:

1. The authors use picrust2 to infer community-level functions from closely related taxa to their 16S rRNA gene data. Since it is already difficult to assign functions to metagenomic data containing information of encoded genes, inferring functions from taxa is even more challenging and may lead to misleading interpretations. Thus, I would advise against using this information as it is highly dependent on the used database and even then, it is not sure if the taxa really encode those functions assigned in the database. On the other hand, I would propose to go more in depth regarding the assigned taxa. For example, test on OTU/ASV level which taxa are distinct for the communities (down to species level). If one can find differences here for certain keystone organisms one could carefully discuss their metabolic potential. Here, I agree with the authors that investigating the functions is needed for answering the questions they stated.

2. I appreciate that the authors combine highly distinct ecosystems in their analysis. Yet, it is worrying that separate protocols (e.g. primers) were used for the different samples. It definitely needs to be addressed how they want to overcome the possibility of a batch effect. Especially since it seems that the one sampling point sampled differently contains different taxa.

Detailed comments:

Line 82: “29.975 ha”. Please transform this in a SI unit.

Line 106: Could the authors please specify if the water in section III, i.e. their sampling point “mangroves”, is only fed by water coming from the other sampling points. Otherwise, one would need to reconsider if it is really a “continuum” as stated in the title.

Line 114: While I appreciate that the authors used replicates in their study, I would propose to specify a bit more how the samples were taken. For example, How high was the water depth. How much distance was between the sampling points at each site (approximately). I did not find the Physical-chemical parameters in their Supplementary tables. Additionally, please specify when the samples were taken since you mention different sampling campaigns for mangrove vs. river and valley.

Line 124: Since a microbiome can change quite quickly after sampling, additional information regarding the transportation time without cooling would be necessary.

Line 131: “bacterial 16S”. Please, see above.

Line 132-138: Since different primers are used for different samples, please report the sequences of the primers in the supplementary.

Line 141: Please, state the used parameters for trimming.

Line 145: What were the clustering settings, e.g. the sequence identity? Were all sequences from all sites clustered together, or separately?

Line 146: It is important to state exactly what tests were used with which settings.

Line 162-163: That would be part of the discussion.

Line 159-170: I was not able find the analysis of functions in the supplementary file S1.

Line 188: Since the authors used different sequencing approaches for the sites, it should be tested if the difference of mangroves is caused by biological or technical reasons.

Line 196: This figure is not easy to interpret. I would suggest to use a heatmap for example to display all taxa in the same plot.

Line 206: Please elaborate how the permanova was used. What does the distance relate to in figure 2B?

Validity of the findings

General:
1. Since the studied ecosystem is well suited for ecology studies it is not surprising that also other studies investigated the Juliana River basin (e.g. https://doi.org/10.7717/peerj.12229). However, it would be necessary to clearly separate the tackled knowledge gaps to the existing studies.

Additional comments

The authors mention the source-valley-estuary form a continuum. Yet, from the shown map it seems to be that the mangrove sampling point is not directly impacted by the other sampling points being another tributary. Since the authors know the sampling point the best, I would like to know their opinion on the impact the upstream water has on the mangrove sampling point.

Reviewer 2 ·

Basic reporting

I have completed reviewing the manuscript entitled “Changes in the ecology of prokaryotes along the continuum source-valley-estuary of a pristine river system in the Brazilian Atlantic Forest” as submitted for consideration in the journal “PeerJ”. This manuscript reports the bacterial and archaeal diversity of stream and mangrove sediments from the Juliana River system, which drains a protected forested catchment and represents an important refuge of biodiversity. As per review criteria, I do not assess impact or novelty.

The reporting, data, and interpretation in the manuscript evaluation summary is, respectively: The research question is well defined and justified, and the reporting is mostly clear, with a few phrases that need better definition. Specifically,

L28-L30: I do not understand the intention of: “A microbiome, however, does not mirror the large-scale prokaryotic patterns” so cannot give specific advice on the wording: please clarify. Furthermore, I suggest deleting “a homeostatic” in L28, this suggests that the paper will dig into feedbacks between structure and function, which it does not.

L64: instead of “foreign groups”, say “invasive populations”? (if that is the intent of the statement)

L69-70 and throughout- the lack of gaps between paragraphs and alternative use of tabs made this sometimes hard to read. Maybe this happened during the processing of the document, but please be careful to help and not confuse the reader’s eye.

L70: “Atlantic ocean” instead of “Atlantic coast”?

L101 (specifically, Figure 1): The map needs a scale bar for distance, and would benefit from the context of showing some relief, tributaries, and/or vegetation coverage as possible.

L122: to the YSI product information, add state and country to be consistent with L129

L131, L136, and throughout: never say just “16S”, and do not say “16S rRNA” when the DNA/gene is the target molecule assayed. 16S rRNA refers to the ribosome itself, not the gene. Change to “16S rRNA gene” here and throughout the manuscript, there are several other instances where this needs to be corrected.

L137: Change to “(Caporaso et aol. 2011), followed by”
L138: delete “Gregory”

L189 (specifically, Figure 2E): Can you please put the data in frame "E" in the same order as the other figures (Source, Valley, Mangrove)? This figure is confusing otherwise.

L209-211: Were these positive or negative relationships? The direction matters for interpretation.

L218: taxa at what level of resolution? Phylum, genus, somewhere in between?

L232-235: vitamin synthesis and aromatic degradation are quite ecologically specific functions, not general metabolic functions
L242: the very specific function of stxB requires a citation.

L257: delete “, our results suggest that”; replace “is” with “could be”

L259: Do the references cited investigate changes in sediment or water microbiomes? I would expect the different microbial habitats to differ in control and pattern.

L289: singular of phyla is phylum
L290: “some negative effect of salinity”

L302: “sediment” instead of “soil”

L313-315: Maybe cite Bergey’s Manual or a textbook instead?

Experimental design

Data collection and methods description follows standard procedures and is close to complete, but requires more detail on the sequence data processing and depth, and there are some results details that could benefit from clarification. Specifically,

L142-143: “For the mangrove samples…” How were source/valley sites merged? Bioinformatics must be explained for all data, and also, all data should be treated the same way; it can otherwise be called into more serious question that the mangrove sample primers are different.

L146: The rarefaction depth must be reported.

L159: PiCrust provides weak functional inference, at best, for microbiomes outside of human gut associated habitat. See https://doi.org/10.1186/s40168-020-00815-y. The functional inference in this report should remain very conserative, and should be accompanied by a clearly stated caveat that most stream and river sediment microorganisms have no cultured representatives or validated and annotated genomes against which to accurately predict functional potential.

L183: The total number of ASVs must be reported.

L202: say “as strongly” instead of “statistically”
L204: delete “significantly”.
P = 0.05 and 0.08 are similar probabilities https://doi.org/10.1080/00031305.2018.1527253

L205-205 and Figure 2B: statistics were based on PERMANOVA, but what metric is on the y-axis? Is it a mean of the Bray-Curtis difference of each e.g. spring site and all the valley and mangrove sites? Specify in the figure and/or figure legend, and text.

L207: instead of “such differences were not found to be significant”, say “there were no differences”.

L209: Is the ordination a PCA or NMDS? The figure says NMDS, which is the more appropriate type of model for 16S rRNA gene community data.

L209-210: instead of “levels of Cu consisted with the most significant”, say “Cu concentration was the most strongly correlated”.

L226 (specifically, Figure 3): This figure is not very useful. The KO abbreviations in the figure must be defined- the reader cannot interpret the figure or results without the full names. The figure legend repeats information in the figure itself but does not provide additional useful information.

L229-230: These are all relative gene counts and cannot be inferred to levels of actual phenotypic potential. It may be possible to infer higher functional redundancy, as the discussion works towards, but only if the number of different genotypes within each KO category is higher in Source samples than in Mangrove samples. The latter analysis is not included in the results, however, and should be if any conclusions about functional redundancy are drawn.

Validity of the findings

Interpretation and context is similarly mostly sound, however some statements are too speculative or poorly defined as written, particularly in the functional interpretation of the data. Specifically,

L261-267: This is too vague a statement. I see some mechanisms are elaborated upon further in the discussion, but this text still needs to include directionality and relationality in every statement. Decreasing diversities with what environmental parameter? How did temperature, salinity, and the trophic status affect diversity?

L280-281: A citation is needed to support the stated difference in diversity of plants.
L281: instead of “guaranteeing different sources”, say “supplying different qualities”

L283: correlated positively or negatively?

L286: Interpretation about nutrient levels or trophic state cannot be made based on abundance of one phylum. Stream microbial composition is least likely to respond to nutrients than other environmental parameters (https://doi.org/10.3389/fmicb.2015.00454), and the default expectation of an undeveloped watershed would be for inorganic nutrient levels to be low, especially in the headwaters (https://doi.org/10.1038/415416a). This interpretation is a particularly big stretch without any water or sediment nutrient data to back it up.

L319-L323: Crenarchaea is a huge and diverse phylum, and includes many important “cold Crenarchaotes” (such as the ammonia oxidizing archaea). Ecological inference about the entire phylum is not possible.

L331-332: As with PiCrust, references from a human associated or clinical context should not be used to interpret data from very different environments.

L348: A quick search shows that Bathyarchaeia have high metabolic diversity (https://www.science.org/doi/10.1126/sciadv.adf5069)

L351: “the predicted metabolism associated” – which predicted metabolism?

L360: delete “, in part because of the higher diversity of nutrient sources in the area” because there is no evidence or citation to support this statement.

---

## Round 0.2 · Major Revisions

Please accept my apologies that the review of your revised manuscript has taken much longer than expected. Unfortunately, one of the reviewers was unable to give it immediate attention, and I wanted to avoid soliciting a new expert. Now I have received the assessments of both original reviewers, who find the manuscript improved. They raise a few additional minor points, which I would like to ask you to consider and respond point by point how you have dealt with them. More import, please revise the manuscript to address the chief concern by Reviewer #2 that your “interpretation of higher diversity resulting from development is insufficient.” Please see his review below for further explanation. Once this issue by Reviewer #2 has been settled, I will have another close look at the manuscript. I may have some final requests at that stage.

Reviewer 1 ·

Basic reporting

The authors improved the language, reference style and background information. If a few changes are made, I think this manuscript can be published (see below).

Raw data is shared.
One thing which remains is that I would encourage the authors to improve the captions of the figures.

Experimental design

Since the code is published this is fine.

Validity of the findings

'no comment'

Additional comments

Line 25-28: I would suggest to merge the two sentences.

Line 157: Maybe change to: “Here, we discuss the diversity…”

Line 159: maybe just add the information that the source is in the mountains in the bracket. -> (a continuum Source-Valley-Estuary). Comparison was carried out from river Source in the mountains to Valley to Mangrove). Then, the following sentence could be deleted.

Line 313-315: Maybe combine these two sentences to:
The Atlantic Forest, being supported by watersheds feeding into the Atlantic Ocean, is one of the most biologically diverse and one of the most vulnerable biomes in the world.

Line 452: “(<30,000)” Please add unit.

Line 479: “is the source of the river in this study” -> “and is designated as the source of the river in this study.”

Line 753-755: “Figure 3A, B, C shows volcano plot summaries of these results (Supplemental File 6). A Venn diagram of these is available as Supplemental Figure 6. “ I would propose to move this to the results part. Same is true for Line 638-639.

Line 737: Somewhere here I would propose to add a new section header for statistical analysis.

Regarding figures: Please add information on number of replicates, significance values, and a short main message to take from the plot. Ideally, the reader would be able to understand the figure with just the caption.
Fig1.: Maybe add information on geochemical parameters -> Which site has high Cu or Zn.

Line 1409: Is this the total phylum name: “Sva0485”. A quick search showed that it is a clade within the proteobacteria.

Reviewer 2 ·

Basic reporting

I have completed reviewing the revised manuscript entitled “Changes in the ecology of prokaryotes along the continuum source-valley-estuary of a pristine river system in the Brazilian Atlantic Forest” as resubmitted for consideration in the journal “PeerJ”. This manuscript reports the bacterial and archaeal diversity of stream and mangrove sediments from the Juliana River system, which drains a protected forested catchment and represents an important refuge of biodiversity. As per review criteria, I do not assess impact or novelty. The reporting, data, and interpretation in the manuscript evaluation summary is, respectively: The research question is well defined and justified, and the reporting is clear, with the revision including acceptable clarification of some previously vague statements.

Experimental design

Data collection and methods description follows standard procedures and is now also standardized among sites, and includes acceptable detail on the sequence data processing and depth.

Validity of the findings

Clarification of results detail has been completed in this revision. Interpretation and context is also more sound, without the speculative functional interpretation of the data; also the authors now qualify their interpretations with the caveat that more data would provide more inferential power, which is fair.

Notably, the authors changed one of the study sites in this revised manuscript from that reported in the original submission. This was a surprising choice. I do understand the rationale given by the authors. After some consideration, I must assume that these data are as good if not better quality than the original, and indeed more appropriate to compare with the other datasets collected using the same methods. While the site is less “pristine” than the original, it is still representative of the ecosystem type, and therefore my opinion is that the singular interpretation of higher diversity resulting from development is insufficient. I request that, when interpreting the higher diversity of the mangrove samples, the authors give attention to the fact that the mangrove system is tidal, therefore experiences a range of salinity and oxygen conditions daily, as well as a mixture of microorganisms from upstream as well as marine origins. This variation is at least as likely as urbanization to affect diversity; in fact, it could also be argued that impacts of development could decrease diversity. And certainly, the observation of halotolerant and sulfur-cycling taxa points to the influence of marine waters and/or source microorgansims on the community.

Overall, the entire study reports on a region for which little information is published, and is valuable for that reason. After a more thoroughly considered interpretation of diversity changes over the continuum of samples, this can be acceptable for publication.

Additional comments

Some specific comments follow:

L108: not sure this sentence is necessary? Or, if required by the university, it may fit better in the acknowledgements section or another location in the manuscript text.

L215-216: suggest minor edit for grammar “as well as Halobacterota which was just below the 2% cutoff”

L220: “Combined, ” (please add comma)

L227: “microbiomes” instead of “metagenomes”
L228: “core” instead of “basal”
L275: “was” instead of “consisted with”

L279-281: the PCA axis represents differences in community composition among samples, direction on the axis is not particularly meaningful unless the difference among samples is also defined. Suggest saying something like “This pattern reflects a different community composition and lower Cu concentration in the Source samples relative to the Valley and Mangrove samples” instead of the current sentence.

L289: “core” instead of “base”
L413: “core” instead of “basal”

---

## Round 0.3 · Minor Revisions

Thank you for addressing the additional comments of the two original reviewers of your manuscript “Ecology of prokaryotes along the continuum source-valley-estuary of a river system in the Brazilian Atlantic Forest." I have not returned the revised manuscript to the reviewers this time but I edited it with a quite heavy hand and would like to ask you to consider these edits as well as a number of queries I have written in the margin of the manuscript.

Some of the changes concern aspects of the manuscript structure (e.g. some rearrangements, deletions and additions), but most are meant to sharpen and streamline the paper. I will try to attach a PDF file with my editorial changes and comments, but will also send you the edited Word file in a separate email for your convenience.

Please systematically remove Methods and Results from the figure and table legends, including in the supplementary materials. There is also scope to further improve the technical quality of the figures. This includes details like the fact that Pielou is a proper name and must therefore be capitalized.

I look forward to hearing from you in due course.

---

## Round 0.4 · accepted · Accept

Thank you for your last thoughtful revision of your manuscript. I only have a few small remaining editorial changes for you to consider (please see the attached PDF files), which I think you can address when you receive the manuscript from the production department. Thank you for submitting your work to PeerJ and for your patience.